# The Role of Epithelial-to-Mesenchymal Transition Transcription Factors (EMT-TFs) in Acute Myeloid Leukemia Progression

**DOI:** 10.3390/biomedicines12081915

**Published:** 2024-08-21

**Authors:** Diego Cuevas, Roberto Amigo, Adolfo Agurto, Adan Andreu Heredia, Catherine Guzmán, Antonia Recabal-Beyer, Valentina González-Pecchi, Teresa Caprile, Jody J. Haigh, Carlos Farkas

**Affiliations:** 1Laboratorio de Investigación en Ciencias Biomédicas, Departamento de Ciencias Básicas y Morfología, Facultad de Medicina, Universidad Católica de la Santísima Concepción, Concepción 4030000, Chile; diego.a21cuevas@gmail.com (D.C.); adolfoagurto@gmail.com (A.A.); aandreu@ucsc.cl (A.A.H.); cguzman@ucsc.cl (C.G.); vgonzalez@ucsc.cl (V.G.-P.); 2Laboratorio de Regulación Transcripcional, Departamento de Bioquímica y Biología Molecular, Facultad de Ciencias Biológicas, Universidad de Concepción, Concepción 4030000, Chile; robertoamigob@gmail.com; 3Departamento de Biología Celular, Facultad de Ciencias Biológicas, Universidad de Concepción, Concepción 4030000, Chile; antorecabal@udec.cl; 4Paul Albrechtsen Research Institute, CancerCare Manitoba, Winnipeg, MB R3E 0V9, Canada; 5Department of Pharmacology and Therapeutics, Rady Faculty of Health Sciences, University of Manitoba, Winnipeg, MB R3T 2N2, Canada

**Keywords:** AML classification, MLL-AF9 fusion, hematopoietic stem cells, epithelial–mesenchymal transition (EMT), genetic aberrations in AML, therapy-resistant AML, extramedullary engraftment

## Abstract

Acute myeloid leukemia (AML) is a diverse malignancy originating from myeloid progenitor cells, with significant genetic and clinical variability. Modern classification systems like those from the World Health Organization (WHO) and European LeukemiaNet use immunophenotyping, molecular genetics, and clinical features to categorize AML subtypes. This classification highlights crucial genetic markers such as FLT3, NPM1 mutations, and MLL-AF9 fusion, which are essential for prognosis and directing targeted therapies. The MLL-AF9 fusion protein is often linked with therapy-resistant AML, highlighting the risk of relapse due to standard chemotherapeutic regimes. In this sense, factors like the ZEB, SNAI, and TWIST gene families, known for their roles in epithelial–mesenchymal transition (EMT) and cancer metastasis, also regulate hematopoiesis and may serve as effective therapeutic targets in AML. These genes contribute to cell proliferation, differentiation, and extramedullary hematopoiesis, suggesting new possibilities for treatment. Advancing our understanding of the molecular mechanisms that promote AML, especially how the bone marrow microenvironment affects invasion and drug resistance, is crucial. This comprehensive insight into the molecular and environmental interactions in AML emphasizes the need for ongoing research and more effective treatments.

## 1. Definition of Acute Myeloid Leukemia (AML), Genetic Variability, and Classification

Acute Myeloid Leukemia (AML) is characterized by the inhibition of proper blood cell differentiation, with a significant blockage at the level of hematopoietic stem cells (HSCs) or early myeloid progenitors like myeloblasts. This leads to rapid disease progression and is clinically diagnosed as AML when there are 20% of AML blasts in the peripheral blood [1]. In contrast, Acute Lymphocytic Leukemia (ALL) arises from a block at the lymphoid progenitors and predominantly affects young children [2]. Chronic leukemias (CLs), on the other hand, allow for more mature and functional blood cells to accumulate [3,4]. AML is more prevalent in elderly adults and accounts for approximately 80% of all leukemias in adults [5], often leading to aberrant hematopoiesis and bone marrow failure. Recent treatments have improved cure rates to around 15% in 60-year-old patients and approximately 40% in those under 60 [6]. In 2018, the global cancer surveillance system (GLOBOCAN) reported a significant incidence of AML worldwide, with 474,519 cases globally and 67,784 in North America, reflecting a prevalence of about 11 cases per 100,000 population [7].

Acute myeloid leukemia (AML) was initially classified from M0 to M7 based on the French–American–British (FAB) system, emphasizing morphological and immunological characteristics [8]. These subtypes include M0 with minimal differentiation and M1 with a higher proportion of mature myeloid forms, which shows advanced stages of cell maturation and is associated with AML1 and ETO fusion proteins [9,10]. M3, or Acute Promyelocytic Leukemia (APL), comprises about 10% of all AML cases and features the PML-RARα fusion protein from the t(15;17) translocation, with a favorable prognosis due to effective treatments with all-trans retinoic acid (ATRA) and arsenic trioxide (ATO) [11,12,13,14]. M4 represents acute myelomonocytic leukemia [15,16], while M5, or monocytic leukemia, often presents with poor prognosis, extramedullary disease, and abnormalities on chromosome 11q, including the *MLLT3* (MLL-AF9) fusion protein, causing the Mixed-Lineage Leukemia (MLL) subtype [17,18,19,20]. Subtype M6, also known as erythroleukemia, is a rare form of AML (comprising less than 5% of AML cases) [21,22,23]. Finally, the rare M7 subtype, megakaryocytic/megakaryoblastic leukemia, exhibits poor differentiation and prognosis [24,25].

The advent of flow cytometry and next-generation sequencing (NGS) has enhanced the understanding of AML’s molecular and genetic complexities, highlighting the need for a more precise classification system to better stratify patients based on genetic profiles, affecting treatment and prognosis [26,27,28]. These classifications are further illustrated in Table 1.

The 2022 International Consensus Classification (ICC) and the World Health Organization (WHO) fifth edition classification updated and introduced new AML categories to align more closely with genetic and clinical data. Notably, the ICC introduced a category for AML with *TP53* mutations, characterized by a poor prognosis due to at least 20% blasts and a *TP53* variant allele fraction over 10% [29,30]. This classification acknowledges the unique biology of *TP53* mutations in AML. Additionally, the ICC has refined the category of AML with myelodysplasia-related changes into two separate groups as follows: one for gene mutations and another for cytogenetic abnormalities, both requiring at least 20% blasts [29,30]. Recent updates from WHO-ICC 2022 and the European Leukemia Net (ELN) guidelines in 2022 reflect these changes, emphasizing the role of genetic and molecular data in AML management, indicative of a shift towards precision medicine in oncology [31,32]. The ELN 2022 updates also introduced a categorization system that considers hierarchical genetic abnormalities, establishing a new category for MDS/AML with 10–19% blasts if specific genetic abnormalities are present, highlighting the importance of genetic markers [32].

Table 2 Summarizes the findings of the WHO 2022 and European LeukemiaNet 2022 guidelines alongside additional literature, detailing the impact of genetic variations on the prognosis of different AML subtypes. AML subtypes are characterized based on specific genetic abnormalities and their corresponding blast percentage requirements, highlighting how these factors influence ELN risk classification.

Next, we will explore one of the well-studied genetic aberrations in AML, specifically within the M5 subtype—the MLL-AF9 fusion protein. We will review the molecular mechanisms underlying its pathogenicity and its significant role in therapy-related AML (t-AML), underscoring how these genetic fusions influence disease progression and impact treatment strategies.

## 2. MLL-AF9 Fusion Protein Oncogenic Mechanisms and Incidence in AML

MLL, a large protein weighing 431 kDa, is encoded by the *KMT2A* gene located on chromosome 11q23 [52,53]. This protein undergoes intracytoplasmic cleavage by the enzyme Taspase 1, resulting in two functional subunits, MLL-N and MLL-C, which are essential for its role in the MLL complex along with WDR5 (WD repeat-containing protein 5), RBBP5 (RB binding protein 5, histone lysine methyltransferase complex subunit), and ASH2L (ASH2-like, histone lysine methyltransferase complex subunit) proteins [54]. This complex is integral to maintaining proper chromatin structure and facilitates the efficient transcription of critical developmental genes, including those involved in hematopoiesis [55,56,57,58]. One of the crucial functions of the MLL complex is the methylation of lysine 4 on histone H3 (H3K4), an epigenetic mark that is vital for the activation of *HOX* genes. These genes are essential for developmental processes and are particularly notable for their roles in maintaining the properties of hematopoietic stem or progenitor cells [56,59]. The epidemiology of *MLL* rearrangements indicates a high incidence in infant leukemias and a significant presence in adult AML, often leading to monocytic differentiation corresponding to FAB classifications AML-M4 or AML-M5. These rearrangements, particularly the MLL-AF9 fusion protein resulting from the t(9;11)(p22;q23) translocation, are associated with a poorer prognosis in AML patients, highlighting the clinical importance of recognizing this genetic alteration for targeted treatment strategies [60,61,62,63,64,65].

The AF9 protein, encoded by the *MLLT3* gene, is a protein varying in molecular weight between 63 and 88 kDa. Part of the YEATS family, the *MLLT3* gene encodes intrinsically disordered proteins and is characterized by its distinctive YEATS domain. AF9 is pivotal in hematopoiesis, primarily tasked with maintaining the population of hematopoietic stem or progenitor cells (HSPCs), essential for the generation and regulation of blood cells [66,67]. AF9 also plays a critical role in gene expression regulation through its interaction with acetylated lysine 27 on histone H3. This interaction is important for enhancing the recruitment of the Like Histone Lysine Methyltransferase protein, encoded by the *DOT1L* gene. Nuclear magnetic resonance (NMR) studies have shown that DOT1L binds to AF9 across three of its domains, facilitating this process [68]; through this interaction, DOT1L specifically targets lysine 79 residues on histone H3 for methylation. This mark plays a pivotal role in chromatin remodeling, leading to the decompaction of the chromatin structure and facilitating the transition from a more condensed heterochromatin state to a more relaxed euchromatin state, promoting gene expression [69].

In the context of AML, the interaction between AF9 and MLL proteins is crucial for the regulation of gene expression, specifically facilitating the binding of the MLL enzyme complex to gene promoters in an active transcriptional state [70]. This interaction significantly promotes the activation of *HOX* genes, which are key transcription factors regulating the development of the anteroposterior axis across various organisms. Their continuous expression is vital for maintaining the undifferentiated state of progenitor cells, similar to blast cells in AML [70].

*HOX* genes, organized into four clusters (*HOXA*, *HOXB*, *HOXC*, *HOXD*), play crucial roles in development and disease. In early vertebrate development, their expression is regulated by chromosomal positioning, with specific clusters being condensed and inaccessible to transcription machinery, thereby inhibiting expression during early phases [42,71,72,73]. These genes are typically downregulated post-embryogenesis and can become aberrantly reactivated in neoplastic conditions, potentially leading to states that favor uncontrolled cellular proliferation [74]. Humans possess 39 *HOX* genes across seven families, with specific genes like *HOXB3*, *HOXB4*, and *HOXA1* to *HOXA10* linked to adverse outcomes in diseases such as acute myeloid leukemia (AML) [75,76,77,78,79,80,81,82].

Specifically, the *RUNX1* and MLL-AF9 interaction disrupts normal hematopoietic gene expression, highlighting the significance of the *NPM1* gene in AML pathology [83]. NPM1 mutations, present in about 30% of AML cases, correlate with *HOX* gene expression and impact leukemogenesis through pathways like the CEBPα pathway, which activates CTBP transcriptional regulators and affects the expression of *HOXA5*, *HOXB5*, and *HOXA10* in *NPM1*-mutant AMLs [83,84,85,86,87].

Therefore, the recognition of the presence of the MLL-AF9 fusion gene in AML is clinically significant, as this genetic alteration can influence treatment selection and therapeutic response monitoring. Patients with this gene fusion may require more aggressive therapeutic approaches and combined treatment strategies to enhance outcomes and overcome resistance to conventional treatments [88].

## 3. First-Line Treatments for AML May Cause t(9;11)—A Mechanistic Perspective

A range of anticancer agents specifically target topoisomerase II (TOP2), an essential enzyme in DNA replication encoded by the *TOP2A* gene, has been used in the treatment of AML. These drugs disrupt the TOP2 catalytic cycle in various steps (Figure 1), leading to an accumulation of TOP2-DNA cleavage complexes and double-strand breaks (DSBs), culminating with cell death. TOP2 poison drugs such as etoposide, teniposide, and doxorubicin inhibit the re-ligation of DNA following TOP2-induced cleavage, while other compounds like quinolone CP-115953 and Azatoxins initiate DNA break formation [89,90]. Notably, doxorubicin, a well-known anthracycline, acts as an inhibitor of DNA re-ligation at lower concentrations (<1 μM) but may interfere with TOP2’s DNA binding at higher concentrations (>10 μM) [90]. Despite their effectiveness, these TOP2 poisons, which remain in clinical use, are linked to severe adverse effects, including the emergence of secondary malignancies [91,92,93]. Notably, treatment-related myelodysplastic syndromes (t-MDSs) often culminate in therapy-related acute myelocytic leukemia (t-AML), presenting significant clinical challenges [94,95]. Etoposide, a chemotherapy drug, is linked to an increased risk of t-AML and t-MDS, particularly when used in cumulative doses exceeding 2000 mg/m^2^/day in the treatment of testicular and extragonadal germ cell tumors [96,97]. The risk is notably higher within the first five years post-treatment, although cases of t-AML/MDS have been reported even two decades after initial treatment with this drug [98]. Systematic reviews reported that in the context of germ cell tumor treatment, the incidence rates of t-AML/MDS far surpass those of spontaneously occurring AML/MDS, underscoring the significant risk increase attributable to etoposide-based chemotherapy regimens [99,100].

Comparative analysis reveals that the risk of developing therapy-related AML/MDS (t-AML/MDS) is significantly higher than that of de novo AML/MDS, with t-AML/MDS being 13 to 200 times more likely to occur. This heightened risk, notably following etoposide and anthracycline treatments, extends to other chemotherapies for various cancers [98,100,101]. A common feature in t-AML is chromosomal translocations, particularly involving the MLL gene on chromosome 11q23, with translocations such as MLL-AF9 being pivotal in leukemia development [102,103,104,105]. The *MLL* gene frequently translocates with partners such as ENL (*MLLT1*), AF4 (*MLLT2*), and AF9 (*MLLT3*), among over a hundred identified partner genes [20,106,107]. These translocations play a crucial role in leukemia development; for instance, MLL-AF9 can transform hematopoietic precursors and induce leukemia in animal models [108,109,110].

The most frequent translocation partners for the *MLL* gene are AF4 (*MLLT2*), AF9 (*MLLT3*), ENL (*MLLT1*), AF10 (*MLLT10*), and the ELL gene [20,106,107]. However, in the case of therapy-induced leukemia, the most frequent translocation partners are AF9, ELL, AF4, and ENL proteins. Interestingly, AF9 is very common in pediatric AML cases with most of the *MLL* translocation at intron 9; however, in adult AML cases with MLL-AF9 fusion protein, MLL translocations occur more frequently at intron 11 [106], probably caused by poison-induced DNA cleavage [19,106].

Taken together, first-line treatments for AML, and other malignancies, use drugs against TOP2 because of their ability to disrupt DNA replication processes. However, their use is associated with significant adverse effects, notably the increased risk of therapy-related malignancies like t-AML and t-MDS, largely because of the induction of chromosomal translocations involving the MLL gene, among others. These insights underscore the need for a delicate balance in cancer therapy, weighing the benefits of effective cancer treatment against the potential for long-term genetic consequences and the development of secondary malignancies. In t-AML, the most frequent translocation presented corresponds to MLL-AF9, and patients with this fusion protein usually have aggressive AML with a bad prognosis. Also, as previously described, once MLL-AF9 has been formed, different target genes and cellular processes will be affected by this fusion protein. This leads to patients having aggressive AML with a bad prognosis, which underscores the need to have a better understanding of the mechanism downstream and upstream of this fusion protein.

**Figure 1 biomedicines-12-01915-f001:**
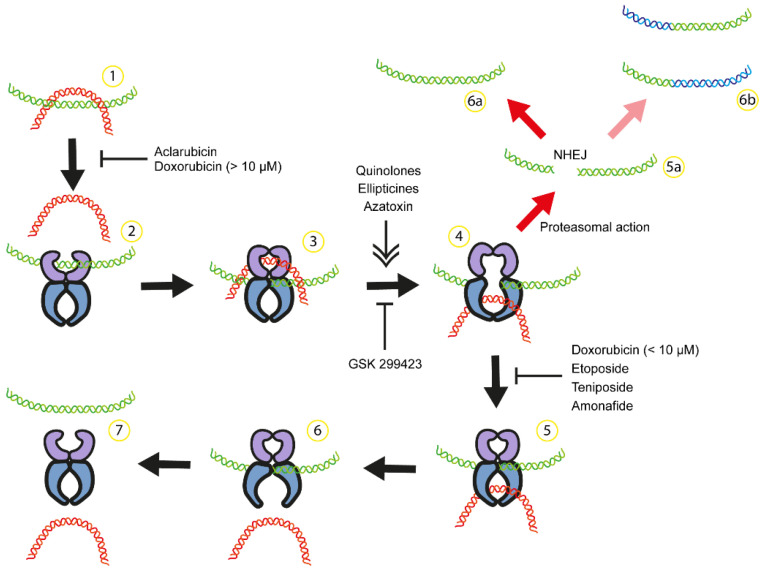
Anticancer drugs can interfere with the catalytic cycle of TOP2, causing chromosomal translocation. (1) DNA supercoiling and catenation. (2) TOP2 dimer binds to one DNA double helix (green), and some compounds can inhibit TOP2 binding to DNA [111]. (3,4) Top2 generates a double-strand break in green DNA in the presence of Mg^2+^, and TOP2 remains attached to both DNA ends. A second DNA double helix (red) passes through the break in an ATP-dependent process. Some compounds can stimulate or inhibit DNA break formation [111]. (5–7) After the red DNA passage is completed, green DNA is re-ligated and both DNAs are released from the enzyme. (5a) Etoposide and doxorubicin can inhibit DNA re-ligation [111], resulting in the accumulation of TOP2 attached to DNA ends. After proteasomal action, DNA with double-strand breaks is repaired by Non-Homologous End-Joining (NHEJ), potentially leading to mutation (6a) or chromosome translocation (6b) [19].

Metabolic reprogramming, including the dysregulation of lipid metabolism, is indispensable for cancer cell survival and propagation through the accumulation of Lipid Droplets (LDs) [112]. LDs not only function as energy reservoirs but also provide building blocks for membrane biosynthesis and produce signaling molecules critical for tumor progression. In AML, elevated LD accumulation leads to poor prognosis and chemoresistance by activating oncogenic pathways and through interaction with the tumor microenvironment [113]. Targeting LD biogenesis has been gaining increased attention, and several inhibitors have shown promise in AML such as hydrophobic aminopeptidase inhibitor CHR2863 plus Rapamycin [114], 3-Methyladenine [115], and Pioglitazone [116], among others, that can complement current AML treatment regimes.

## 4. Emergence of Epithelial-to-Mesenchymal Transition (EMT) Factors in the Risk and Progression of AML: The Role of ZEB Transcription Factors

The acknowledgment of increased t-AML incidence following first-line treatments for AML underscores a critical need for innovative therapeutic strategies that mitigate the risk of secondary malignancies while effectively combating primary disease states. As research progresses, a promising area of exploration involves the epithelial–mesenchymal transition (EMT) factors, known for their pivotal roles in cell differentiation and migration. In the context of AML, EMT factors contribute to the plasticity of leukemic cells, influencing their ability to resist apoptosis and evade the immune system, thus presenting a dual challenge and opportunity in leukemia treatment. By targeting these EMT factors, new treatments could potentially disrupt the cellular mechanisms that contribute to the aggressiveness and poor prognosis often observed in MLL-AF9-mediated AML.

Based on the work of Prange et al. [84] and Stavropoulou et al. [117], it is increasingly clear that *ZEB1* and *ZEB2* genes may act as transcriptional targets of MLL fusion proteins. Prange et al. highlighted the genome-wide binding of MLL-AF9 and MLL-AF4 fusion proteins in AML cell lines, revealing both shared and unique target genes marked by specific epigenetic signatures. Their study demonstrated how MLL fusions, alongside subsets of transcription factors, can deregulate critical gene programs in AML, suggesting a role for *ZEB1* and *ZEB2* within this framework. Similarly, Stavropoulou et al. 2016 underscored the impact of cellular origin on AML aggressiveness and identified EMT-related genes, including ZEB1, associated with poor outcomes in AML patients [117]. Their work emphasized the complex interplay between MLL fusion proteins and cellular origin in determining AML characteristics, where *ZEB1* and *ZEB2* emerge as potential mediators in the MLL-driven leukemic processes. These findings offer a refined perspective on the molecular mechanisms underpinning MLL-associated leukemia, highlighting the potential of *ZEB1* and *ZEB2* as EMT factors to be critical components within this oncogenic network in AML.

## 5. Role of ZEB Transcription Factors

*ZEB1/2* transcription factor dysregulation has previously been demonstrated to play pathological roles in the EMT processes involved in (1) the malignant dissemination (metastasis) of epithelial-derived tumor cells [118,119], (2) the acquisition of cancer or tumor stem cell properties [120,121], and (3) the development of treatment resistance [122,123]. The *ZEB* transcription factors are large multidomain proteins that contain both amino-terminal and carboxy-terminal Zinc Finger DNA binding domains that bind to bi-partite E-box binding sites (CACCT(G), sometimes CACANNT(G)) [124] in the promoter/enhancer regions of target genes. These proteins can either suppress transcription by recruiting corepressor complexes such as the Nucleosome Remodeling and Deacetylase (NuRD) complex that contain Histone Deacetylase (HDAC) 1/2, resulting in chromatin closure and gene repression, or enhance transcription by attracting additional transcription complexes containing p300 acetylation factor that opens chromatin, allowing access to transcription factors [125,126] and the basic transcriptional machinery.

*ZEB2* [127,128] and, more recently, *ZEB1* [129,130] have been demonstrated to play important roles in regulating murine hematopoiesis as well as immune cell differentiation and function [131,132,133] distinct from their roles in EMT. In hematopoietic stem and progenitor cells (HSPCs) *ZEB2* and, to a lesser degree, *ZEB1* limit the inappropriate expression of innate and adaptive immune cell programs [129,130]. Upon lineage commitment, *ZEB2* maintains distinct immune programs to produce defined populations of functional macrophage, dendritic, natural killer, and T cells [131,132,133,134]. Not only does *ZEB2* ensure immune cell lineage fidelity that is unique to a given lineage, but it does so with very little overlap in terms of common gene expression programs it regulates. To exemplify this point, *ZEB2* interacts with signals from the tissue environment to specify macrophage identity that is unique to the host organ (lung vs. colon, etc.) with very little overlap of common Differentially Expressed Genes (DEGs) among organs, which is associated with murine *Zeb2* loss [134]. Given the importance of these unique genetic programs in controlling lineage fate/function, it is perhaps not surprising that ZEB protein dysregulation can lead to various different forms of leukemia via lineage-specific mechanisms including myeloid lineage transformation leading to AML [135]. Recently, the role of *ZEB1* in macrophage differentiation [136] and dendritic cell homeostasis [137] has been described, pointing out that oncogenic *ZEB1* can alter these processes during leukemogenesis.

Within the AML context, *ZEB1* and *ZEB2* may become dysregulated through direct transcriptional control of the MLL-AF9 and MLL-AF4 oncofusion proteins [116,117]. Additionally, *ZEB1/2* are known to be negatively regulated by the miR200 family of miRNAs [138], and in their absence, *ZEB* protein levels may accumulate to oncogenic levels. Within AML, the miR200 family of miRNAs has been found to be methylated and repressed, associated with increased *ZEB2* levels [139,140]. Additional AML oncofusions including AML-ETO and PML-RARα have also been demonstrated to transcriptionally upregulate *Zeb2* [141], implying that *ZEB2* upregulation may be a common driver of AML progression.

In terms of the importance of ZEB proteins in driving AML development/progression, *ZEB2* plays an oncogenic role. In two separate and unrelated genetic screening approaches, *ZEB2* was found to be involved in myeloid and lymphoid leukemic transformation and a novel genetic dependency in murine and human AML [120,139]. In the first instance, *ZEB2* was overexpressed from the ROSA26 (R26) safe-harbor locus in Tie2-Cre lineage marked cells, which includes the endothelium and entire hematopoietic system that is derived from the hemogenic endothelium of the Aorta–Gonad–Mesonephros (AGM) region [120]. Tie2-Cre, R26^Tg/Tg^ mice develop spontaneous T cell transformation around 6 months of age that genetically and phenotypically resembles Early Thymic Progenitor Acute Lymphoblastic Leukemia (ETP-ALL). ETP-ALL is characterized by transformed early T cell progenitors that express HSPC as well as myeloid cell markers. On a sensitized p53 null background, Tie-Cre, R26^Tg/Tg^ mice also developed AML and B-ALL but at lower frequencies. These results imply that *ZEB2* may play oncogenic roles in myeloid and B cells, but additional genetic hits may be required for *ZEB2*-mediated leukemic transformation. In line with the latter, genetic alterations frequently implicate *ZEB2* in various translocations and mutations within T-lymphoid leukemia and AML, with 14q32 rearrangements involving the *BCL11B* gene marking a distinct subgroup. These genetic rearrangements form a unique expression profile that significantly affects leukemia biology and patient prognosis [142].

In a separate unbiased CRISPR/Cas9-based screening approach, *ZEB2* was found to be a top genetic dependency involved in both human AML cell lines and MLL-AF9 murine AML proliferation [139]. Knock-down of *ZEB2* in human AML cell lines resulted in enhanced morphological differentiation as assayed by May–Grunwald–Giemsa staining analysis and increased mature CD11B myeloid marker expression in flow cytometry analysis [139]. Genetic deletion of *ZEB2* using a tamoxifen-inducible Cre-based approach in an established murine MLL-AF9 model was found to significantly increase overall survival [130]. Moreover, several studies support the oncogenic role of *ZEB2* in human AML. Mechanistically, it has been shown that miR-454-3p targets *ZEB2*, playing a critical role in AML progression. Overexpression of miR-454-3p induces apoptosis and autophagy in AML cells by downregulating *ZEB2* expression, which concurrently inhibits the AKT/mTOR signaling pathway [143]. Additionally, the overexpression of *ZEB2-AS1* long non-coding RNA (lncRNA), which leads to increased *ZEB2* levels, has been associated with poorer clinical outcomes in acute myeloid leukemia [144]. These findings highlight the necessity of maintaining tight regulation of *ZEB2* to prevent its upregulation to oncogenic levels.

The oncogenic role of *ZEB1* in AML is more controversial, as Almotiri et al. recently postulated that *ZEB1* may act as a tumor suppressor given that murine *Zeb1* deletion in MLL-AF9 models can accelerate AML progression [129]. Bioinformatically, evidence was provided that *ZEB1* levels may be lower in certain subtypes of AML [129]. However, simultaneous deletion of both murine *Zeb2* and *Zeb1* in murine MLL-AF9 AML settings was found to significantly increase the overall survival of mice transplanted with MLL-AF9 secondary tumor cells compared with nontreated vehicle-treated controls [130]. A separate bioinformatic analysis by Almotiri et al. demonstrated that both *ZEB1* and *ZEB2* in human AML are significantly higher in the leukemic blast population than in the bulk tumor population, suggesting that these transcription factors may be diluted in the bulk RNA-sample analysis as well as in overall levels of expression used in the Kaplan–Meier survival curves [130]. Consistent with an oncogenic role of *ZEB1* in AML, Stavropoulou et al. identified murine *Zeb1* as an essential target of MLL-AF9 in HSC-like leukemic AML blast populations, which play an important role in leukemic blast invasion to extramedullary sites [117]. In line with the latter, it has been demonstrated that the hematopoietic transcription factor ZNF521 increases its levels in AML with MLL rearrangements, enhancing hematopoietic stem cell transformation via ZEB1, among other genes [145]. *ZEB1* was also found to play an oncogenic role in human AML cell lines by increasing PI3K/AKT signaling [146], which relies on p53-PTEN pathway modulation. In the latter study, it was demonstrated that *ZEB1* expression is negatively correlated with tumor suppressor P53 expression, and *ZEB1* can directly bind to P53 as a molecular mechanism to exert oncogenesis.

The complex role of *ZEB1* within AML extends to its effect on the immunological landscape, where it downregulates CD8 T cell activity and promotes the expansion of Th17 cells, enhancing the survival and proliferative capabilities of leukemia cells in the *AML* niche. Recently, Bassani et al. demonstrated that high levels of *ZEB1* correlate with increased Th17 cell development and a pro-invasive phenotype associated with poor patient outcomes [147,148]. The analysis of *ZEB1* expression in larger datasets of AML identifies two distinct groups, *ZEB1*^high^ and *ZEB1*^low^, each with specific immunological and gene expression signatures. Importantly, *ZEB1*^high^ patients exhibit increased expression of IL-17, *SOCS2*, and TGF-β (Transforming Growth Factor Beta) pathways and a negative association with overall survival [147].

The role of *ZEB1* is beyond the protein level. It has been shown that mutations in the splicing factor 3b subunit 1 encoded by the *SF3B1* gene are frequent in myelodysplastic neoplasms (MDS). In this context, a strong upregulation of Circular RNAs (circRNAs) processed from the *ZEB1* locus has been reported, which may impact mitochondrial function and cellular metabolism in myelodysplastic syndrome (MDS) [149]. Moreover, *ZEB1* interacts with long non-coding RNA *MALAT1*, influencing its activity and stability through m6A modification and thereby modulating the aggressiveness of AML [150]. Taken together this evidence highly suggests that *ZEB1* serves as an oncogene at various levels of AML.

Overall, the *ZEB1* and *ZEB2* transcription factors play important roles in the progression and aggressiveness of acute myeloid leukemia (AML) by influencing the epithelial–mesenchymal transition, stem cell characteristics, and therapy resistance. These factors are often dysregulated because of transcriptional control by MLL fusion proteins or repression by the miR200 family of miRNAs, leading to enhanced oncogenic activity. Specifically, *ZEB2* has been linked to both myeloid and lymphoid leukemic transformations, affecting hematopoietic differentiation and immune modulation, while *ZEB1* impacts the immune system by regulating T cell activities and promoting Th17 cell expansion, which correlates with poor clinical outcomes. Given their significant roles, targeting ZEB proteins offers a promising therapeutic approach to potentially improve treatment efficacy and reduce the incidence of secondary malignancies like therapy-related AML (t-AML).

At present, small molecules are available with the ability to interact with EMT-TFs to potentially halt AML progression, and chemotherapeutics currently available for AML treatment may be combined with specific inhibitors of EMT-TFs to provide better treatment against AML progression. Currently, three noteworthy studies have utilized ZEB2 inhibitor compounds for this purpose (all commercially available). The first study focused on Honokiol, a phenolic compound primarily investigated in breast and kidney cancer research, which downregulated EMT genes including ZEB2 [151]. The second study employed Teniposide at low doses to inhibit ZEB2, significantly reducing the pulmonary colonization of breast cancer cells [152]. While Teniposide has been used in AML treatment at higher doses, it leads to therapy-related relapse at standard clinical doses; thus, combined therapy is necessary [153]. Importantly, the third study identified the novel small molecule CD3254 as an effective agent for promoting mouse chemical reprogramming (0.5 μM). This molecule operates through the CD3254–RXRα axis to activate the RNA exosome and specifically downregulates Zeb1/2, Twist, and SNAI1 EMT factors [154]. These drugs hold potential for inclusion in AML treatment strategies to inhibit EMT-TFs and improve therapeutic outcomes.

## 6. Role of SNAI Transcription Factors

The SNAI family (*SNAI1*, *SNAI2*, and *SNAI3*) of transcription factors have also been demonstrated to play essential roles in EMT [155] as well as the acquisition of cancer stemness properties [156] and drug resistance [157]. SNAI proteins have conserved carboxy-terminal Zinc Finger DNA binding domains that are also capable of binding to E-box elements (5′-CAGGTG-3′) in the promoter of epithelial genes such as E-cadherin, which is involved in its transcriptional repression. Regarding the *miR200-SNAI1* relationship, studies have shown that *SNAI1* plays a key role in facilitating EMT and mesoderm differentiation during a particular phase of embryonic stem cell differentiation, which aligns with the early epiblast stage. In this period, SNAI1 modifies the levels of various miRNAs, notably those within the miR200 microRNA family, but it is unclear if this relationship occurs in hematopoiesis [158]. The SNAI proteins all have an amino-terminal SNAI-GFI (SNAG) binding domain that is responsible for the recruitment and promoter binding of epigenetic factors such as *KDM1A* (Lysine Demethylase 1A) (LSD1 protein), which is responsible for histone 3 lysine 4 demethylation (H3K4me1/2) and gene repression [159]. This SNAG domain, as its name implies, is shared with other proteins including the GFI1/1B transcription factors that play important roles in hematopoiesis [160]. SNAI proteins have also been demonstrated to play roles in hematopoiesis. *SNAI2* has been demonstrated to act downstream of the c-Kit signaling pathway in HSPCs (hematopoietic stem and progenitor cells) [161] and SNAI2/3 have been demonstrated to play functionally redundant roles in B and T cell development [162].

From an AML perspective, SNAI1 has been found to be overexpressed broadly in both primary AML samples and cell lines irrespective of the driver mutations [163] present. shRNA-mediated knock-down (KD) of SNAI1 in human AML cell lines was found to lead to enhanced morphological differentiation as assayed by May–Grunwald–Giemsa staining analysis and increased mature CD11B (Integrin αM protein) myeloid marker expression in flow cytometry analysis [163], similar to the effects associated with ZEB2 KD [142]. Also, tamoxifen-mediated knockout of murine *Snai1* significantly improved the survival of mice transplanted with MLL-AF9 as well as AML-ETO/N-RAS models of AML [164], also similar to *ZEB2* knockouts [130].

Mechanistically, increased mouse *Snai1* was demonstrated to alter myeloid development and lead to the enhanced self-renewal of myeloid progenitors [163]. Vav-iCre, R26-*Snai1*^Tg^ mice go on to develop a myeloproliferative disorder that progresses to full AML transformation, with 50% of these mice succumbing to AML by 400 days [163]. These effects on myeloid differentiation were demonstrated to be due to binding to *KDM1A* (LSD1), as point mutants in the SNAG (Snail1/GFI) protein domain that ablated LSD1 binding while still preserving protein stability did not alter myeloid development [163]. Also, RNA-seq and ChIP-seq experiments demonstrated that increased *SNAI1* expression leads to enhanced recruitment of LSD1 to repress SNAI1 target genes, potentially at the expense of other key transcriptional regulators that require LSD1 for ensuring normal hematopoiesis such as GFI-1 (Growth Factor Independent 1 Transcriptional Repressor) proteins [163,165].

## 7. LSD1 and Other Potential Therapeutic Targets

The ability of EMT transcription factors to alter LSD1 function is an emerging theme in AML and other leukemias such as ETP-ALL, as *ZEB2* has previously been demonstrated to interact with LSD1 [139,166] in both of these settings. Within the ETP-ALL context, increasing levels of *ZEB2* were shown to be associated with increased susceptibility to LSD1 inhibitors [166]. The same likely holds true for the levels of *SNAI1* and *ZEB2* in human AML. LSD1 inhibitor therapy is an emerging promising new cancer therapy not only in AML [166] but other solid forms of cancer including Non-Small Cell Lung Cancer (NSCLC) and childhood sarcomas [164,167]. Whether or not EMT-TF levels also drive LSD1 inhibitor sensitivity in solid tumor settings remains to be determined.

Like other monotherapies, resistance mechanisms to LSD1 inhibition are likely to occur. As an example, *ZEB2* has been demonstrated to upregulate IL-7R expression, which may be part of its ability to transform T cells [120,168]. Enhanced IL-7 signaling has recently emerged as a resistance mechanism for developing LSD1 inhibitor resistance [169], as the IL-7R pathway drives increased JAK/STAT signaling and increases the amount of the pro-survival BCL2 protein. ZEB2-LSD1 complexes in ETP-ALL were shown to repress pro-apoptotic BIM protein levels. Therefore, combination therapies using LSD1i with JAK/BCL2 inhibition were demonstrated to be synergistic in treating ETP-ALL in vitro and in vivo in patient-derived xenotransplant settings [169]; this combination was demonstrated to shift the balance of survival factors towards pro-apoptotic programs.

IL-7 is not known to be expressed in AML; however, ZEB2 has been demonstrated to modulate other myeloid-relevant cytokines such as IL-6 and G-CSF [128], and SNAI1 may enhance TNF/NFkb signaling [163], which may activate similar pro-survival pathways in AML.

Overall, as we learn more about this crosstalk, the ability of ZEB and SNAI family members to influence key epigenetic modulators such as LSD1 and control key cytokine signaling pathways may offer new therapeutic options for treating AML. The roles of SNAI and ZEB proteins are summarized in Table 2. In addition, the ability of ZEB and SNAI proteins to control genes involved in adhesion and migration that may also offer new therapeutic avenues in AML.

## 8. Role of SNAI2 in AML

It has been shown that *SNAI2*, like *SNAI1*, is closely linked with the progression and treatment response of leukemia. *SNAI2* promotes leukemogenesis, and its loss or pharmacological inhibition impairs leukemic stem cell (LSC) self-renewal and delays leukemia progression. At the transcriptional level, Slc13A3, a direct target of SNAI2 in LSCs, restricts the self-renewal of LSCs and significantly prolongs recipient survival, highlighting its potential as a therapeutic target [170]. Furthermore, SNAI2’s involvement in chemoresistance complicates treatment strategies; its expression is associated with robust resistance to conventional chemotherapy in LSCs, underscoring the need for targeted therapies that can overcome this barrier [171]. Taken together, these studies underscore SNAI2’s oncogenic role in leukemia biology, influencing stem cell dynamics and conferring drug resistance, each aspect offering potential opportunities for therapeutic intervention.

## 9. Role of TWIST1 in AML

*TWIST1*, a transcription factor, is central to the pathophysiology of acute myeloid leukemia (AML), affecting multiple biological processes that govern disease progression and treatment response. *TWIST1* promotes cell growth, drug resistance, and progenitor clonogenic capacities in myeloid leukemia, and it is linked to poor prognostic factors [172]. In line with this, a recent study demonstrated *TWIST1* expression and promoter methylation levels were significantly upregulated in AML tissues and cell lines, and its expression was further downregulated by using demethylating agent 5′-azacitidine (5-Aza)-treated cells, leading to apoptosis [173]. The PI3K/AKT signaling pathway was positively regulated by Twist1, suggesting that Twist1 serves as an oncogene in AML.

Moreover, *TWIST1* is notably involved in the extramedullary manifestations of AML, where it significantly promotes tissue invasion and metastasis. Treatments with *TWIST1*-siRNA or metformin downregulate *TWIST1*, including *SNAI2*, which is associated with significant impairment of migration and invasion processes [174]. *TWIST1* is also essential for the viability and self-renewal of leukemia stem cells (LSCs), especially in MLL-AF9 leukemia, thus promoting disease initiation and maintenance [175]. The role of *TWIST1* is not only restricted to LSCs since it has been demonstrated that *TWIST1* influences bone marrow microenvironment interactions by modulating mesenchymal stem cell differentiation, which in turn promotes leukemia expansion [176]. In line with the latter, the role of *TWIST1* in promoting AML is also seen in the recruitment of regulatory T cells within the tumor microenvironment, potentially providing new targets for immunotherapeutic approaches [177].

TWIST1’s role extends to chemoresistance, where it interacts with DNA methyltransferase 3a (DNMT3a) to regulate resistance to decitabine, a key therapeutic agent in treating AML [178]. This interaction underscores the potential for targeting TWIST1 in therapeutic strategies aimed at overcoming chemoresistance.

In summary, the extensive involvement of *TWIST1* in AML suggests its utility not only as a biomarker for disease progression and treatment response but also as a promising target for therapeutic intervention. This could lead to the development of more personalized and effective treatment strategies for AML, transforming current paradigms and improving patient outcomes [172]. A comparative overview of EMT factors and their roles in AML are presented in Table 3.

## 10. Spread of AML Cells

In this section, we review some of the most studied molecular components of microenvironments responsible for AML spread and extramedullary hematopoiesis. AML, as a liquid tumor, inherently possesses greater mobility and penetration capabilities than solid tumors. Still, AML cells require robust molecular mechanisms for invading other bone marrow sites or establishing Extramedullary Engraftment (hereafter referred to as EME) [179]. Certain molecular factors involved in AML-EME can hinder immunotherapy effectiveness and protect AML cells within the bone marrow [180]. Others enhance engraftment properties, facilitate cell motility, and enable the transition of AML cells between the bone marrow and bloodstream, contributing to EME [181,182].

A critical area of focus is the role of EMT factors in AML-EME, which may elucidate aspects of the disease’s aggressive progression. A recent study using RNA-seq analyses to compare gene expression between AML patients with and without relapse identified EMT-related genes such as *CDH2* [183], *LOX* [184], and *COL3A1* [185,186,187] as strong correlates of AML prognosis and EME. *CDH2*, also known as N-cadherin, plays a pivotal role in cell adhesion and motility, thereby enhancing the potential for leukemic cells to invade distant tissues. Similarly, *LOX* and *COL3A1*, which are crucial to extracellular matrix (ECM) functionality, support the structural dynamics necessary for tumor metastasis and invasion [186].

## 11. Intravasation and Extravasation Mechanisms of AML

The mechanism by which AML cells gain the ability to extravasate from the bone marrow and into the bloodstream marks the beginning of AML-EME. A key component in this process is the formation of invadosomes, cellular structures in cancer cells that degrade the ECM and facilitate entry into the bloodstream [188]. These structures are also crucial during extravasation, where AML cells must breach the endothelial cell barrier to exit the vasculature and invade surrounding tissues. The transformation of AML cell structures into invadopodia, actin-based membrane protrusions capable of degrading the ECM, is critical for penetration through the vasculature endothelium. Invadopodia also recruit ECM proteases, aiding this process [189,190]. These actin-based membrane protrusion structures can degrade the ECM and cause cell penetration through the endothelium of vasculature and, in addition, invadopodia can recruit ECM proteases that contribute to this process [191].

Proteins commonly found in invadopodia include cortactin [192], actin filament nucleating proteins like N-WASP (Neural Wiskott–Aldrich Syndrome protein) [193], scaffold proteins such as Tks4 protein (Tyrosine Kinase Substrate With Four SH3 Domains, *SH3PXD2B* gene) [194], Tks5 (SH3 and PX domains 2A protein, *SH3PXD2A* gene) [194,195], and various metalloproteases [196]. These proteins work together to facilitate cell motility and ECM degradation, enabling intravasation and extravasation in metastatic AML, including other leukemias [197,198]. The Vascular Endothelial Growth Factor (*VEGF* gene) produced by AML cells induces bone marrow degradation, specifically targeting laminin and type IV collagen, and promoting vessel sprouting [199]. This process creates thin ECM “hotspots”, making these sites more susceptible to invasion by AML cells [200]. Once relocated to other bone marrow sites, AML cells can grow by re-establishing their initial niche microenvironments.

From a mechanistic perspective, the Myocardin-Related Transcription Factors and Serum-Response Factor (MRTF-SRF) pathway mediates some migration properties of AML cells and is notably present in the MLL-AF9 model [201]. E-selectin, a surface glycoprotein expressed by the vasculature [202], binds to ligands on both normal immune and AML cells. Activated endothelial cells expressing E-selectin may signal AML cell attachment and facilitate intravasation, working in conjunction with motility strategies [200]. While in the bloodstream, AML cells require additional molecular tools to evade immune detection. *SETDB1* (the SETB1 Protein), a lysine methyltransferase key in epigenetic regulation, helps AML cells escape immune response by methylating retrotransposons [203]. While *SETDB1* can repress tumorigenic genes, it also enables AML cells to evade immune detection [203,204]. Similarly, *CD36* (the CD36 Protein), a multifunctional scavenger receptor, is linked to EME dissemination and increased relapse risk post-chemotherapy. Blocking the CD36 protein delays AML relapse, while binding of thrombospondin 1 (TSP-1, the *THBS1* gene) to CD36 promotes AML migration [205]. Altogether, these findings suggest that a balance is required between molecular properties for migration and those that improve homing and treatment resistance.

During extramedullary invasion, AML cells may adhere to microenvironments near critical organs during their journey through the bloodstream, initiating extramedullary colonization [206]. The survival of AML cells within the bone marrow depends on the chemokine Stromal Cell-Derived Factor (the SDF-1 protein, the *CXCL12* gene), which acts as a survival and attachment factor within the bone [207]. SDF-1 is produced by stromal cells in the spleen, bone marrow, and extramedullary sites like the skin and central nervous system, facilitating AML cell attachment outside the bone marrow [208]. This aspect of AML is less studied than in ALL.

Interestingly, SDF-1 receptor (the *CXCR4* gene) is expressed variably among AML cells [209]. AML cells show lower *CXCR4* expression compared with normal bone marrow cells [210], suggesting that reduced *CXCR4* expression is linked to loss of bone marrow attachment. Blocking *CXCR4* increases AML cell migration, indicating CXCR4’s role in regulating bone marrow niche adherence [182]. Additionally, poor prognosis in AML patients is associated with the expression of *CXCR4* or E-selectin [211]. *ZEB2*, a regulator of *CXCR4*, therefore emerges as a potential target in the EME process [127,139].

Regarding the extravasation process, there is an opportunity for targeted therapies. Extravasation by E-selectin is reduced by the action of Uproleselan, an E-selectin antagonist that also induces AML cell mobilization from the bone marrow into the bloodstream, making them more vulnerable to chemotherapy [212]. On the other hand, Integrin β2, expressed by AML cells, binds to Matrix Metalloprotease 2 (the *MMP2* gene), which is responsible for extramedullary cell invasion and metastasis by degrading the ECM [188]. Altogether these findings highlight the complexity of AML metastasis and suggest novel potential targets for therapeutic intervention. An analysis of the referred molecular factors that significantly impact the behavior of AML cells are presented in Table 4.

AML-EME can manifest as sarcomas, which are solid myeloblasts proliferating outside the bone marrow. This occurs in approximately 2.5–9.1% of adults with AML [214]. However, certain AML subtypes, such as those involving the t(8;21) translocation, show a higher incidence of AML-EME, as reported in 18–24% of cases [215]. Locations reported include soft tissues, ovaries, intestines, testes, breasts, lymph nodes, renal masses, and eyes, but the most common ones are soft tissue and lymph nodes [215]. CD56, a Neural Cell Adhesion Molecule (*NCAM1* gene) commonly expressed in the brain, has been consistently linked to poor prognosis in AML [216,217,218]. The brain, considered a sanctuary tissue, is often associated with a poor leukemia prognosis [218].

The infiltration of AML-EME cells into the central nervous system (CNS), including the skull, meninges, and brain, is considered rare, though its incidence may be underestimated because of infrequent diagnostic procedures [219]. The most documented cases involve cranial bone marrow infiltration by AML cells. For example, a patient case report indicated a bone marrow replacement disorder in the skull’s bone marrow, leading to an AML diagnosis [219]. Notably, CNS involvement is more common in pediatric AML than in adult cases [220]. Key risk factors for CNS involvement in AML include complex karyotypes, AML relapse, FAB M5 classification, high LDH levels (Lactate Dehydrogenase A, LDHA), the presence of other extramedullary AML manifestations, and FLT3-ITD mutations [221]. In contrast, CNS involvement in APL (Acute Promyelocytic Leukemia) typically presents as meningeal leukemia and is more frequently observed [214].

In summary, the role of epithelial–mesenchymal transition (EMT) factors is particularly notable in AML-EME, impacting prognosis and disease progression. Genes like *CDH2*, *LOX*, and *COL3A1* have been identified as key correlates of AML-EME. Also, intravasation and extravasation mechanisms, including the formation of invadosomes and invadopodia, are crucial in the spread of AML cells. These mechanisms are supported by proteins such as cortactin, N-WASP, Tks4, Tks5, and metalloproteases, which facilitate cell motility and degrade the extracellular matrix. Adaptation to new tissue environments requires AML cells to evade immune detection by many of the discussed mechanisms. In conclusion, understanding the molecular mechanisms behind AML’s EME ability is crucial. This knowledge provides insight into potential therapeutic targets and strategies to combat the spread of AML, particularly in challenging cases involving extramedullary sites including CNS involvement.

The overall roles of EMT-TFs in normal hematopoiesis, AML transformation and EME are summarized in Figure 2.

## 12. Overall Conclusions and Future Directions

AML, with its profound genetic, molecular, and clinical heterogeneity, continues to pose significant challenges in oncology. This review explored the complex landscape of AML, considering the current disease classification, molecular characteristics, and the dynamic mechanisms that drive its aggressive progression and spread. The presence of genetic aberrations, such as *FLT3*, *NPM1* mutations, and MLL-AF9 gene fusion, are pivotal in predicting the prognosis, relapse, and therapeutic responses in AML, emphasizing the crucial role of personalized medicine in its management.

The exploration of EMT factors, particularly the *ZEB*, *SNAI*, and *TWIST* gene families, reveals promising avenues for new therapeutic targets. Their significant roles in regulating hematopoiesis and influencing AML’s aggressive EME behaviors highlight potential innovative treatment strategies that could target these pathways. Furthermore, understanding the molecular mechanisms that facilitate AML-EME—including the adaptation of AML cells to various microenvironments and their ability to evade immune surveillance—is essential. This knowledge opens new avenues for research and therapeutic interventions, which are crucial for developing novel strategies to manage and potentially overcome the aggressive nature of this malignancy.

The need to enhance the precision of genetic and molecular diagnostics is crucial for accurately categorizing AML subtypes and predicting treatment responses. Current strategies for stratifying AML increasingly rely on next-generation sequencing (NGS) and other genomic technologies. These methods, including targeted gene panels and low-coverage whole genome sequencing, provide a more efficient classification of leukemia, pushing the boundaries of precision medicine. Such advancements not only improve diagnostic accuracy but also facilitate the development of targeted therapeutic strategies in AML.

Additionally, the development of targeted therapies that address the unique molecular aberrations of each AML subtype is critical. This approach may include novel drug combinations and advanced immunotherapies, including novel gene-editing technologies. Given the significant role of the bone marrow microenvironment in the progression of AML, targeting this niche presents a promising strategy to combat the disease. The exploration of EME and CNS involvement in AML also requires further investigation to develop effective treatments for these particularly challenging manifestations of the disease.

Lastly, AML’s management requires a multifaceted approach that integrates genetics, molecular biology, and innovative therapeutic strategies. Future research and clinical approaches should aim to incorporate these elements, moving towards more personalized and effective treatment modalities for patients afflicted with this complex form of leukemia.

## Figures and Tables

**Figure 2 biomedicines-12-01915-f002:**
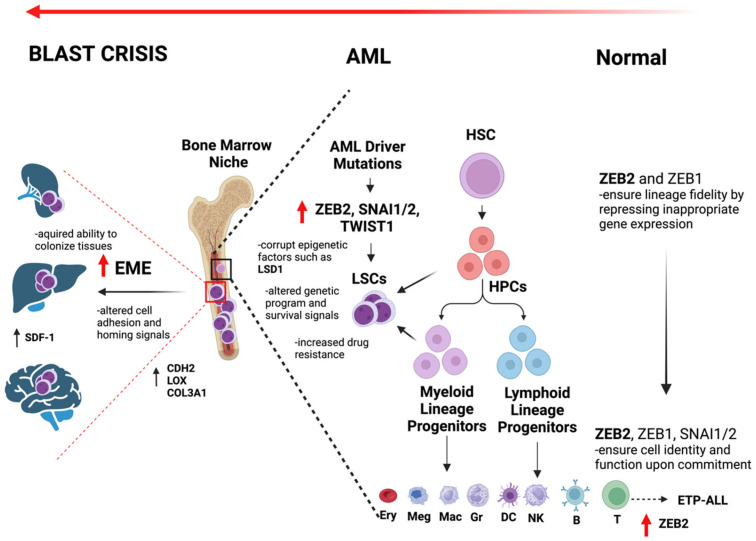
Overview of the role of EMT factors in normal hematopoiesis, AML development, and EME. EMT factors such as *ZEB2* ensure hematopoietic lineage fidelity during early hematopoiesis by restraining mature gene expression programs. Later in hematopoiesis, EMT factors lock in cell identity once committed. Upregulated expression of *ZEB2* can specifically transform T cell lineage and cause Early T cell Precursor Acute Lymphoblastic Leukemia (ETP-ALL). AML driver mutations can increase EMT factor expression, including *ZEB1/2*, *SNAI1/2*, and *TWIST1*, which can corrupt epigenetic factors such as LSD1 and lead to leukemic stem cell (LSC) gene expression program and transformation. EMT factors can also enhance survival signal pathway expression and increase drug resistance. EMT processes also drive extramedullary tissue engraftment, tissue colonization, and blast crisis development by altering AML cell adhesion and homing and survival signals. Ery—erythrocyte, Meg—megakaryocyte, Mac—macrophage, Gr—granulocyte, DC—dendritic cell, NK—natural killer cells, B—B cells, T—T cells, EME—extramedullary engraftment, HSC—hematopoietic stem cell, HPC—hematopoietic progenitor cell, LSC—leukemic stem cell.

**Table 1 biomedicines-12-01915-t001:** Classification and characteristics of acute myeloid leukemia (AML) subtypes according to the French–American–British (FAB) system.

Subtype	Characteristics	Prognosis and Features
M0	High percentage of minimally differentiated blasts, negative for peroxidase, confirmed with myeloid markers by flow cytometry.	Poor prognosis, associated with complex chromosomal abnormalities.
M1	Less than 10% promyelocytes, blasts lack granules with distinct nucleoli, >3% positive for myeloperoxidase.	
M2	Presence of mature cells, including cells with Auer rods, associated with t(8;21) translocations and AML1-ETO/ETO-AML1 fusion proteins.	More favorable prognosis when linked with specific translocations.
M3 (APL)	Hypergranulated promyelocytes, bilobed nuclei, characterized by the PML-RARα fusion from t(15;17) translocation. Treatable with ATRA and arsenic trioxide (ATO).	Favorable prognosis due to effective treatment options.
M4	Presence of monocytes and promonocytes in bone marrow. The M4Eo variant shows abnormal eosinophils and inv(16) cytogenetic abnormality.	Favorable prognosis for variants like M4Eo depending on WBC levels.
M5	High number of monocytic lineage cells, over 30% blast cells in BM or PB, often associated with 11q abnormalities and MLL gene rearrangements (e.g., MLL-AF9).	Poor prognosis, common extramedullary disease, and linkage to severe clinical features.
M6	Over 50% nucleated erythroid cells in bone marrow with substantial abnormalities, positive for PAS staining and glycophorin A.	Rare, less than 5% of AML cases, associated with severe developmental abnormalities in erythroid cells.
M7	Poor megakaryocytic differentiation, megakaryoblasts with scant cytoplasm and dense chromatin, negative for common stains, confirmed by CD41 and electron microscopy.	Extremely rare (~1% of cases), poor prognosis due to aggressive nature and difficulty in treatment due to morphological variability in the cells.

**Table 2 biomedicines-12-01915-t002:** Subtypes of AML according to specific genetic markers and the required percentage of blasts. Each subtype is further classified by the ELN risk category, ranging from favorable to adverse. The presence of certain genetic abnormalities such as t(15;17)/PML:RARA, RUNX1:RUNX1T1, and mutations in NPM1 or TP53 influence the prognosis and therapeutic approach. References are provided for each subtype to support evidence.

AML Type	Blast %	Genetics	ELN Risk Class (2022)	Literature (2024)	Refs.
APL with t(15;17)/PML::RARA	>10%	*PML:RARA*	-	Favorable	[33,34]
APL with other RARA rearrangements	>10%	Various *RARA* rearrangements	-	Variable, depends on the rearrangement	[35]
AML with t(8;21)/RUNX1::RUNX1T1	>10%	*RUNX1:RUNX1T1*	Favorable	Favorable	[36]
AML with inv(16) or t(16;16)/CBFB::MYH11	>10%	*CBFB:MYH11*	Favorable	Favorable	[37,38]
AML with t(9;11)/MLLT3::KMT2A	>10%	*MLLT3:KMT2A*	Intermediate	Intermediate	[39]
AML with other KMT2A rearrangements	>10%	Various *KMT2A* rearrangements	-	Variable, depends on the rearrangement	[39]
AML with t(6;9)/DEK::NUP214	>10%	*DEK:NUP214*	Adverse	Adverse	[40]
AML with inv(3) or t(3;3)/GATA2; MECOM	>10%	*GATA2; MECOM*	Adverse	Adverse	[41]
AML with other MECOM rearrangements	>10%	Various *MECOM* rearrangements	-	Adverse	[42]
AML with other rare recurring translocations	>10%	Rare recurring translocations	-	Adverse	[43]
AML with t(9;22)/BCR::ABL1	>10%	*BCR:ABL1*	Adverse	Adverse	[44]
AML with mutated NPM1	>10%	Mutated *NPM1*	Favorable	Favorable	[45]
AML with bZIP CEBPA mutations	>10%	bZIP *CEBPA* mutations	Favorable	Favorable	[46]
AML/MDS with mutated TP53	10–19%/>20%	Mutated *TP53*	Adverse	Adverse	[47]
AML/MDS with myelodysplasia-related gene mutations	10–19%/>20%	Myelodysplasia-related gene mutations	-	Adverse	[48]
AML with myelodysplasia-related cytogenetic abnormalities	10–19%/>20%	Myelodysplasia-related cytogenetic abnormalities	-	Intermediate	[49]
AML not otherwise specified (NOS)	10–19%/>20%	-	-	-	[50]
Myeloid sarcoma	Not specified	-	Adverse	Adverse	[51]
MDS with mutated TP53	0–9%	Multi-hit *TP53* mutation or *TP53* mutation (VAF > 10%) and complex karyotype often with loss of 17p	Adverse	Adverse	[29,30]
MDS/AML with mutated TP53	10–19%	Any somatic *TP53* mutation (VAF > 10%)	Adverse	Adverse	[29,30]
AML with mutated TP53	>20%	Any somatic *TP53* mutation (VAF > 10%)	Adverse	Adverse	[29,30]

**Table 3 biomedicines-12-01915-t003:** Comparative overview of EMT factors and their roles in AML: summary of the diverse roles of the *ZEB1*, *ZEB2*, *SNAI1*, *SNAI2*, and *TWIST1* epithelial–mesenchymal transition (EMT) factors, namely, across various dimensions relevant to acute myeloid leukemia (AML) pathophysiology. Each column represents a specific EMT factor, outlining its involvement in EMT processes, roles in hematopoiesis, regulation by the miR200 family of miRNAs, influence on AML patient outcomes, interactions with oncofusion proteins, findings from genetic screenings, functional roles in immune cell differentiation, contribution to leukemic transformation, and potential as therapeutic targets. The table incorporates references to significant studies, providing a broad yet detailed perspective on the molecular and cellular functions of these factors in AML and highlighting their potential impacts on disease progression and treatment outcomes.

Feature	ZEB1	ZEB2	SNAI1	SNAI2	TWIST1
Roles in EMT Processes	Involved in malignant dissemination and metastasis [116,117]	Plays a role in cancer or tumor stem cell properties, development, and treatment resistance [118,119]	Essential for EMT, cancer stemness, and drug resistance [155,156,157]	Promotes leukemogenesis and influences chemotherapy resistance [170,171]	Central to AML pathophysiology; affects growth and drug resistance [172]
Roles in Hematopoiesis	Lesser degree of influence compared with ZEB2 [129,130]	Limits inappropriate expression of immune cell programs [131,132,133,134]	Influences stem and progenitor cell functions [158]	Impairs LSC self-renewal, restricts LSC self-renewal via Slc13a3 [170]	Impacts progenitor clonogenic capacities [175]
Regulation by the MiR200 Family of miRNAs	Negatively regulated, lower levels in certain AML subtypes [138]	Negatively regulated; absence leads to oncogenic levels [138,139,140]	Relationship in hematopoiesis is unclear [158]	Not specified	Not specified
Influence on AML Patient Outcomes	Associated with poor outcomes, essential for leukemic blast invasion [117,130]	Upregulation associated with leukemic blasts [130]	Overexpression contributes to impaired differentiation and enhanced self-renewal [163]	Associated with poor clinical outcomes [171]	Linked to poor prognostic factors; promotes tissue invasion [172,174]
Oncofusion Protein Interactions	Upregulated by MLL-AF9 and MLL-AF4 [117]	Upregulated by AML-ETO, MLL-AF9, MLL-AF4, and PML-RARα [116,117]	Not clear	Not specified	Notably involved in extramedullary manifestations [174]
Genetic Screening Findings	Deletion may accelerate AML progression [129,130]	Involved in myeloid and lymphoid leukemic transformation [120,139]	Knock-down enhances morphological differentiation and improves survival [163]	Not specified	Essential for viability and self-renewal of LSCs [175]
Functional Roles in Immune Cell Differentiation	Plays a role in macrophage differentiation [136] and dendritic cell homeostasis [137]	Ensures immune cell lineage fidelity [131,132,133,134]	Implicated in myeloid development and self-renewal of progenitors [161,162]	Not specified	Influences bone marrow microenvironment interactions [176]
Contribution to Leukemic Transformation	Potentially oncogenic, may act as a tumor suppressor [129,130]	Involved in myeloid leukemia transformation [135]	Leads to myeloproliferative disorders and AML transformation [162,163]	Promotes leukemogenesis [170]	Promotes disease initiation and maintenance [175]
Potential Therapeutic Targets	Could offer novel approaches for AML treatment if targeted [146,147,148]	Inhibition may improve outcomes [139,140,141,169]	Knockout or inhibition improves survival [163]	Targeting could impair LSC self-renewal and chemoresistance [171]	Targeting TWIST1 could overcome chemoresistance and influence treatment [178]

**Table 4 biomedicines-12-01915-t004:** An analysis of various molecular factors that significantly impact the behavior of AML cells, particularly in the context of metastasis. The factors are evaluated based on their influence in three key areas including survival, motility, and adherence.

Factor	Survival	Motility	Adherence
SDF-1	[207]		[208]
METTL-3	[213]		
Integrin β		[188]	[188]
N-WASP		[193]	
Tks4, Tks5		[194,195]	
E-selectin		[202]	[202]

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
