# Peer review of "The Role of Epithelial-to-Mesenchymal Transition Transcription Factors (EMT-TFs) in Acute Myeloid Leukemia Progression"

_biomedicines, 2024, doi:10.3390/biomedicines12081915_

Round 1

Reviewer 1 Report

Comments and Suggestions for Authors

Estimated Authors,

I've read with great interest the present narrative review about AML and the biology of transcription factors involved into the pathogenesis of this disease.

From my point of view, the paper is substantially well written, but several recommendations could be shared with the Authors in order to improve it further.

For one, according to the main title, the objective of this review should be the "role of epithelial to mesenchymal transition transcription factors (EMT-TFs) in disease progression". In fact, the first sections of the paper discuss in very extensive terms the role of genetics of AML. From my point of view, sections 2,3,4 and mostly 5 should be simplified as not consistent with the main core of the study.

Second, some minor recommendations:

- please double check the notation of genes and proteins, as the capital letters are not consistently used across the main text;

- please provide in every table / figure the meaning of all acronyms and abbreviations (e.g. AML, acute myeloid leukemia and so on)

Author Response

“Estimated Authors,

I've read with great interest the present narrative review about AML and the biology of transcription factors involved into the pathogenesis of this disease.

From my point of view, the paper is substantially well written, but several recommendations could be shared with the Authors in order to improve it further.

For one, according to the main title, the objective of this review should be the "role of epithelial to mesenchymal transition transcription factors (EMT-TFs) in disease progression". In fact, the first sections of the paper discuss in very extensive terms the role of genetics of AML. From my point of view, sections 2,3,4 and mostly 5 should be simplified as not consistent with the main core of the study.”

R: We thank this reviewer for the comments. Sections 1-3 have been simplified to better align with the main objective of the review. This suggestion helps to maintain a focused discussion on the role of epithelial to mesenchymal transition transcription factors (EMT-TFs) in AML progression and potential drug use in AML. The review sections now read as follows:

  1. Definition of Acute Myeloid Leukemia (AML), genetic variability, and classification
  2. MLL-AF9 fusion protein oncogenic mechanisms and incidence in AML
  3. First-line treatments for AML may cause T(9;11): A mechanistic perspective
  4. Emergence of epithelial to mesenchymal transition (EMT) factors in the risk and progression of AML: Role of ZEB transcription factors
  5. Role of ZEB transcription factors
  6. Role of SNAI transcription factors
  7. LSD1 and other potential therapeutic targets
  8. Role of SNAI2 in AML
  9. Role of TWIST1 in AML
  10. Spread of AML cells
  11. Intravasation and extravasation mechanisms of AML
  12. Overall conclusions and future directions

As per this reviewer's suggestion, we changed the title from “Deciphering the Complexities of Acute Myeloid Leukemia: Emerging Roles of Epithelial to Mesenchymal Transition Transcription Factors (EMT-TFs) in Disease Progression” to “The Role of Epithelial to Mesenchymal Transition Transcription Factors (EMT-TFs) in Acute Myeloid Leukemia Progression.”

“Second, some minor recommendations:

- please double check the notation of genes and proteins, as the capital letters are not consistently used across the main text;

- please provide in every table / figure the meaning of all acronyms and abbreviations (e.g. AML, acute myeloid leukemia and so on).”

R: The notation of genes and proteins has been thoroughly reviewed and corrected across the text to ensure consistency with capital letters as appropriate. According to the HUGO guidelines, human gene symbols have been kept in all italicized capital letters and the corresponding proteins in capital letters, while for mouse genes, only the first letter is capitalized. Additionally, a comprehensive review of all acronyms and abbreviations has been conducted throughout the text. Their meanings are now provided at their first occurrence, and a list of abbreviations has been added. Also, an abbreviations list is provided in the Abbreviatures section towards the end of the review.

Reviewer 2 Report

Comments and Suggestions for Authors

The manuscript is a review article focusing on various aspects of acute myeloid leukemia, AML. There is a definition and classification of AML. After summarizing types of AML based on the genetic aberrations, the authors focus on one type of AML with MLL-AF9 fusion. Further, additional parameters and features related to AML are discussed. 

Specific points:

1. Kindly spell all the abbreviations, at least ones when first introduced. I addition, a list of Abbreviations can be added. For example, it could be spelled and explained what are MLL, KMT2A, and all other abbreviations, especially not commonly and broadly used ones. Also, WDR5, RbBP5, and ASH2L; HOX, AF9, MLLT3, TGF, BFGF, etc., etc.

2. A table summarizing types of AML, classification, from M0 to M7, as described in the text, could be also useful to supplement the text, for better visualization (optional). E.g., a simple table with types of AML and corresponding features. 

3. Line 196. "acomulation". Kindly check.

4. Line 250. "Mg+2". Is it "Mg2+ "?

5. The lines 711-714 could be removed because they are not relevant to this article: "Institutional Review Board Statement: Not Applicable. Informed Consent Statement: Not Applicable. Data Availability Statement: Not Applicable. Acknowledgments: Not Applicable. " This information is optional and the lines can be removed if "not applicable" is the choice.

Comments on the Quality of English Language

Spell abbreviations; proofread to fix the typos

Author Response

“The manuscript is a review article focusing on various aspects of acute myeloid leukemia, AML. There is a definition and classification of AML. After summarizing types of AML based on the genetic aberrations, the authors focus on one type of AML with MLL-AF9 fusion. Further, additional parameters and features related to AML are discussed.

Specific points:

  1. Kindly spell all the abbreviations, at least ones when first introduced. I addition, a list of Abbreviations can be added. For example, it could be spelled and explained what are MLL, KMT2A, and all other abbreviations, especially not commonly and broadly used ones. Also, WDR5, RbBP5, and ASH2L; HOX, AF9, MLLT3, TGF, BFGF, etc., etc.”

R: We thank this reviewer for this suggestion, as requested by reviewer #1, we included all abbreviations throughout the text and as a separate section at the end of the text.

“2. A table summarizing types of AML, classification, from M0 to M7, as described in the text, could be also useful to supplement the text, for better visualization (optional). E.g., a simple table with types of AML and corresponding features.”

R: We have added a new table (Table 1) to the main text summarizing the types of AML using the FAB classification from M0 to M7 to clarify the differences with the former Table 1 (now Table 2). This table depicts the subtype, some characteristics and prognosis/ associated features of each subtype of AML. Please see line 76 of the revised manuscript.

“3. Line 196. "acomulation". Kindly check.”

R: The typo has been corrected to "accumulation".

“4. Line 250. "Mg+2". Is it "Mg2+ "?”

R: The notation has been corrected to Mg2+

“5. The lines 711-714 could be removed because they are not relevant to this article: "Institutional Review Board Statement: Not Applicable. Informed Consent Statement: Not Applicable. Data Availability Statement: Not Applicable. Acknowledgments: Not Applicable. " This information is optional and the lines can be removed if "not applicable" is the choice.”

R: The lines mentioning Institutional Review Board Statement, Informed Consent Statement, Data Availability Statement, and Acknowledgments have been removed as suggested.

Reviewer 3 Report

Comments and Suggestions for Authors

The paper offers a thorough and up-to-date review of AML, covering its genetic diversity, classification systems, and potential therapeutic targets. It is well-structured and provides valuable insights into the molecular mechanisms of AML. However, to enhance its impact, the following revisions are recommended:

  1. Condense Redundant Sections: Streamline the discussion on FAB subtypes to maintain focus and avoid redundancy.
  2. Deepen Analysis: Provide a deeper analysis of specific genetic markers and their clinical implications to strengthen the paper's contribution to AML research.
  3. Empirical Support: Include more empirical evidence or preliminary data to support the speculative elements on new therapeutic targets.

Overall, the paper is a valuable contribution to the field of AML research and, with minor revisions, is suitable for publication.

Comments on the Quality of English Language

Minor changes in English language must be done.

Author Response

The paper offers a thorough and up-to-date review of AML, covering its genetic diversity, classification systems, and potential therapeutic targets. It is well-structured and provides valuable insights into the molecular mechanisms of AML. However, to enhance its impact, the following revisions are recommended:

Condense Redundant Sections: Streamline the discussion on FAB subtypes to maintain focus and avoid redundancy.”

R: We thank the reviewer for this suggestion. This aligns with the previous reviewers' inquiries, and we have condensed the former sections 1-6 into one, as suggested by Reviewer #1 as well.

“Deepen Analysis: Provide a deeper analysis of specific genetic markers and their clinical implications to strengthen the paper's contribution to AML research.”

R: As requested by previous reviewers, we condensed sections 1-6 of the manuscript to streamline the discussion of EMT factors in AML. Additionally, we added a new table discussing the FAB clinical classification of AML in comparison with the updated clinical classification of AML. This approach provides a streamlined yet comprehensive discussion of the specific genetic markers in AML.

“Empirical Support: Include more empirical evidence or preliminary data to support the speculative elements on new therapeutic targets.”

R: We thank the reviewer for this suggestion. As clinical implications, we now discuss metabolic reprogramming and potential drug targeting based on metabolic reprogramming in AML (see lines 256-265 of the new text). Additionally, we have included a new section on various treatments using small molecule inhibitors associated with ZEB transcription factors in EMT, as also suggested by reviewer #4 (lines 435-450). Furthermore, section #7 now discusses the use of LSD1 inhibitors in AML (lines 498-530).

“Overall, the paper is a valuable contribution to the field of AML research and, with minor revisions, is suitable for publication.”

R: We thank this reviewer for this commentary.

Reviewer 4 Report

Comments and Suggestions for Authors

The Review "Deciphering the Complexities of Acute Myeloid Leukemia:  Emerging roles of epithelial to mesenchymal transition transcription factors (EMT-TFs) in disease progression" by Diego Cuevas, Roberto Amigo, Adolfo Agurto, Adan Andreu Heredia, Catherine Guzmán, Antonia Recabal-Beyer, Valentina González-Pecchi, Teresa Caprile, Jody J. Haigh  and Carlos Farkas"  explores the complex landscape of AML with particular regard to EMT factors, particularly the ZEB, SNAI and TWIST gene families, as promising avenues for new therapeutic targets. The review is well-written and contains novel insights; however, it could be further improved.

Firstly, I suggest re-modulating the structure.

The first three paragraphs could be merged into a single paragraph with subsections. Likewise, the discussion of the various transcription factors could also be merged.

In the paragraph 4 the authors could discuss the role of lipid droplets in acute myeloid leukemia, with particular reference to AML MA9. This aspect is important because lipid droplets represent a target that could assist in anti-leukemic therapy.

On line 257, close the parenthesis.

Recently, a reciprocal feedback loop between MA9 and Znf521 was described, which is able to sustain leukemia progression. The result of this interaction is also the activation of Zeb1. The authors could include this aspect in the review, in the paragraph 6.

The authors could summarize the final message of the review in a graphical abstract.

At present, are there small molecules able to target epithelial-to-mesenchymal transition transcription factors (EMT-TFs) in AML progression.

The current chemotherapeutics used in AML treatment could be combined with specific inhibitors of epithelial-to-mesenchymal transition transcription factors (EMT-TFs) to address AML progression?

Author Response

“The Review "Deciphering the Complexities of Acute Myeloid Leukemia:  Emerging roles of epithelial to mesenchymal transition transcription factors (EMT-TFs) in disease progression" by Diego Cuevas, Roberto Amigo, Adolfo Agurto, Adan Andreu Heredia, Catherine Guzmán, Antonia Recabal-Beyer, Valentina González-Pecchi, Teresa Caprile, Jody J. Haigh  and Carlos Farkas"  explores the complex landscape of AML with particular regard to EMT factors, particularly the ZEB, SNAI and TWIST gene families, as promising avenues for new therapeutic targets. The review is well-written and contains novel insights; however, it could be further improved.

Firstly, I suggest re-modulating the structure.

The first three paragraphs could be merged into a single paragraph with subsections. Likewise, the discussion of the various transcription factors could also be merged.”

R: We thank this reviewer for this suggestion. Aligning with the previous reviewers, the first three sections have been merged into a single section for better clarity and to streamline the discussion on EMT transcription factors. Please see lines 38-114 of the new manuscript

“In the paragraph 4 the authors could discuss the role of lipid droplets in acute myeloid leukemia, with particular reference to AML MA9. This aspect is important because lipid droplets represent a target that could assist in anti-leukemic therapy.”

R: We appreciate the reviewer's suggestion to include a discussion on the role of lipid droplets in acute myeloid leukemia (AML), particularly with reference to MLL-AF9 AML. In response, we have added a section on metabolic reprogramming, highlighting the dysregulation of lipid metabolism as critical for cancer cell survival and propagation. Specifically, we discuss how lipid droplets (LDs) function as energy reservoirs, provide building blocks for membrane biosynthesis, and produce signaling molecules essential for tumor progression. In AML, elevated LD accumulation is associated with poor prognosis and chemoresistance due to the activation of oncogenic pathways and interactions with the tumor microenvironment. Furthermore, we detail several inhibitors targeting LD biogenesis, such as Hydrophobic aminopeptidase inhibitor CHR2863 plus Rapamycin, 3-Methyladenine, and Pioglitazone, which show promise in complementing current AML treatment regimes. This new section can be found in lines 256-265 of the revised manuscript with the correspondent references.

“On line 257, close the parenthesis.”

R: The parenthesis has been closed.

“Recently, a reciprocal feedback loop between MA9 and Znf521 was described, which is able to sustain leukemia progression. The result of this interaction is also the activation of Zeb1. The authors could include this aspect in the review, in the paragraph 6.”

R: R: We thank the reviewer for highlighting this. In response, we have included a discussion on this aspect in section 5 of the revised manuscript. Specifically, we have noted that the hematopoietic transcription factor ZNF521 increases its levels in AML with MLL rearrangements, enhancing hematopoietic stem cell transformation via ZEB1, among other genes. This addition can be found between lines 392-396 of the new review.

“The authors could summarize the final message of the review in a graphical abstract.”

R: We consider figure 2 as a graphical abstract to provide an overview of the role of EMT factors in normal hematopoiesis, AML development, and EME. For us, this figure effectively summarizes the final message of the review.

“At present, are there small molecules able to target epithelial-to-mesenchymal transition transcription factors (EMT-TFs) in AML progression.

The current chemotherapeutics used in AML treatment could be combined with specific inhibitors of epithelial-to-mesenchymal transition transcription factors (EMT-TFs) to address AML progression?”

R: In response to the reviewer's inquiry, we have added a new section discussing the potential of small molecules to target EMT-TFs in AML progression. Specifically, we mention that current chemotherapeutics for AML may be combined with specific inhibitors of EMT-TFs to enhance treatment efficacy. We highlight three studies: the first utilizing Honokiol to downregulate EMT genes, including ZEB2; the second employing low doses of Teniposide to inhibit ZEB2; and the third identifying CD3254 as an effective agent to downregulate Zeb1/2, Twist, and SNAI1 EMT factors. These studies indicate that combining these inhibitors with current AML treatments could potentially improve therapeutic outcomes. Please see lines 434-450 of the new manuscript.

Round 2

Reviewer 3 Report

Comments and Suggestions for Authors

After revising the article, the authors have introduced the suggested changes, improving it considerably, and the article has been accepted for publication in the final version. 

Comments on the Quality of English Language

Minor changes in English language must be done.